# *Kalanchoe tomentosa*: Phytochemical Profiling, and Evaluation of Its Biological Activities *In Vitro*, *In Vivo*, and *In Silico*

**DOI:** 10.3390/ph17081051

**Published:** 2024-08-09

**Authors:** Jorge L. Mejía-Méndez, Gildardo Sánchez-Ante, Yulianna Minutti-Calva, Karen Schürenkämper-Carrillo, Diego E. Navarro-López, Ricardo E. Buendía-Corona, Ma. del Carmen Ángeles González-Chávez, Angélica Lizeth Sánchez-López, J. Daniel Lozada-Ramírez, Eugenio Sánchez-Arreola, Edgar R. López-Mena

**Affiliations:** 1Programa de Edafología, Colegio de Postgraduados, Campus Montecillo, Carr. México Texcoco km 36.4, Montecillo 56230, Mexico; jorge.mejiamz@udlap.mx (J.L.M.-M.); carmeng@colpos.mx (M.d.C.Á.G.-C.); 2Departamento de Ciencias Químico-Biológicas, Universidad de las Américas Puebla, Ex Hacienda Sta. Catarina Mártir S/N, San Andrés Cholula 72810, Mexico; yulianna.minuttica@udlap.mx (Y.M.-C.); karen.schurenkamperco@udlap.mx (K.S.-C.); ricardo.buendiaca@udlap.mx (R.E.B.-C.); 3Tecnologico de Monterrey, Escuela de Ingeniería y Ciencias, Av. Gral. Ramón Corona No 2514, Colonia Nuevo México, Zapopan 45121, Jalisco, Mexico; gildardo.sanchez@tec.mx (G.S.-A.); diegonl@tec.mx (D.E.N.-L.); alsl@tec.mx (A.L.S.-L.)

**Keywords:** Crassulaceae, genus *Kalanchoe*, *Kalanchoe tomentosa*, biological properties, phytochemistry

## Abstract

In this work, the leaves of *K. tomentosa* were macerated with hexane, chloroform, and methanol, respectively. The phytochemical profiles of hexane and chloroform extracts were unveiled using GC/MS, whereas the chemical composition of the methanol extract was analyzed using UPLC/MS/MS. The antibacterial activity of extracts was determined against gram-positive and gram-negative strains through the minimal inhibitory concentration assay, and *in silico* studies were implemented to analyze the interaction of phytoconstituents with bacterial peptides. The antioxidant property of extracts was assessed by evaluating their capacity to scavenge DPPH, ABTS, and H_2_O_2_ radicals. The toxicity of the extracts was recorded against *Artemia salina* nauplii and *Caenorhabditis elegans* nematodes. Results demonstrate that the hexane and chloroform extracts contain phytosterols, triterpenes, and fatty acids, whereas the methanol extract possesses glycosidic derivatives of quercetin and kaempferol together with sesquiterpene lactones. The antibacterial performance of extracts against the cultured strains was appraised as weak due to their MIC_90_ values (>500 μg/mL). As antioxidants, treatment with extracts *executed* high and moderate antioxidant activities within the range of 50–300 μg/mL. Extracts did not decrease the viability of *A. salina*, but they exerted a high toxic effect against *C. elegans* during exposure to treatment. Through *in silico* modeling, it was recorded that the flavonoids contained in the methanol extract can hamper the interaction of the NAM/NAG peptide, which is of great interest since it determines the formation of the peptide wall of gram-positive bacteria. This study reports for the first time the biological activities and phytochemical content of extracts from *K. tomentosa* and proposes a possible antibacterial mechanism of glycosidic derivatives of flavonoids against gram-positive bacteria.

## 1. Introduction

Infections caused by bacteria represent a major challenge for public human health care and the global economy. Pathogenic bacteria can be transmitted through various mechanisms (e.g., airborne, droplet, or vehicular) and cause urinary tract infections, pneumonia, meningitis, and superficial wound infections [1]. Epidemiologically, the resistance of bacteria to current antimicrobials is known as antimicrobial resistance (AR), and it has been associated with more than ~1.27 million deaths worldwide [2].

Under healthy physiological conditions, oxidative stress is associated with the promotion of immune responses, modulation of cell signaling pathways, muscle contraction, hormone secretion, and neurotransmission [3,4,5]. When altered by excessive alcohol consumption, unbalanced diet, smoking, chronic diseases, and overexposure to environmental factors, such as aromatic hydrocarbons and phthalates [6], it can contribute to the initiation and development of cardiovascular (e.g., atherosclerosis, hypertension, and myocardial infraction [7], neurodegenerative (e.g., amyotrophic lateral sclerosis, and Alzheimer’s and Parkinson’s disease) [8], metabolic (e.g., diabetes, nonalcoholic fatty liver disease, and obesity) [9], cancer, and chronic inflammatory (e.g., asthma, rheumatoid arthritis, and bowel disease) disorders [10].

When investigating traditional sources of bioactive compounds, there are various approaches by which their activity can be evaluated; some of them include *in vitro*, *in vivo*, and *in silico* models. In vitro, assays are required for screening the possible biological activities of molecules, aiding in the selection of the most promising candidates, identifying the main bioactive constituents, and assessing the need for separation and characterization methods [11]. In contrast, *in vivo* techniques are necessary to validate the therapeutic performance of molecules by considering their interactions, safety, and toxicity [12]. Given the fact that the implementation of *in vitro* and *in vivo* assays can be time-consuming and expensive, *in silico* analyses can enable the rapid evaluation of potential bioactive compounds by considering their interaction and affinity with overexpressed receptors or essential protein moieties involved in pathological events [13,14].

Traditional medicinal plants constitute an attractive alternative to treat or prevent diseases. Depending on the presence of bioactive compounds in their organs, medicinal plants can exert various biological activities, such as antimicrobial, anticancer, antioxidant, and anti-inflammatory [15]. The family Crassulaceae is the seventh largest family in regions of Southern Africa, as it comprises 35 genera and ~1500 species of succulent herbs and small shrubs [16]. The representative genera of the family Crassulaceae are *Adromischus* Lem., *Cotyledon* L., *Kalanchoe* Adans., and *Tylecodon* Tölken [17]. In contrast to other genera, the genus *Kalanchoe* includes a variety of succulent perennial plants that are used to prepare extracts, plant juices, or cataplasms and, herein, apply to treat infectious, musculoskeletal, respiratory, genitourinary, gastrointestinal, and carcinogenic disorders *in vitro* and *in vivo* [18,19].

The biological activities of species from the genus *Kalanchoe* are due to the presence of bioactive secondary metabolites such as bufadienolides (e.g., bryotoxin, bersaldegenin, and bufalin), flavonoids (e.g., derivatives of eupafolin, patuletin, and quercetin), phenols (e.g., vanillic acid, sinapic acid, and gallic acid), tocopherols (e.g., α-, β-, δ-, and γ-tocopherol), terpenes (e.g., amyrin, bryophynol, and friedelin), and steroids (e.g., hydroxycholesterol, bryophyllol, and stigmasterol) [19]. Some species where these molecules have been reported are *K. pinnata* [20], *K. diagremontiana* [21], *K. blossfeldiana* [22], *K. tubiflora* [23], and *K. gracilis* [24].

*K. tomentosa*, also known as panda plant or chocolate soldier, is a succulent plant with dense white hair in its leaves that is predominantly distributed in Madagascar, but its presence has also been reported in regions from Central and South America [25,26]. The scientific evidence that validates the therapeutic use of this species is limited; for example, recent studies demonstrated that the dichloromethane extract from the leaves of *K. tomentosa* can exert bacteriostatic activity against *Staphylococcus aureus* and *Klebsiella pneumoniae*. In the same report, the antibacterial activity of the obtained extract was related to the presence of β-sitosterol, which was identified by UV-vis, thin-layer chromatography (TLC), and nuclear magnetic resonance (NMR) analyses [27].

In the same regard, it was unveiled that methylene chloride, ethanol, and *n*-butanol extracts from the same species contain triterpenoids such as friedelin, phytosterols such as phytol, β-sitosterol, stigmasta-7,25-dien-3β-ol, or flavonoids such as eriodictyol, isovitexin, quercetin, and kaempferol-3-*O*-rutinoside. These compounds were separated by TLC and identified by UV-vis, NMR spectroscopy, and positive ion electron ionization mass spectrometry [28]. The identified natural products were associated with the recorded antimicrobial, cytotoxic, and antioxidant activities of the extracts [28]. On the other hand, treatment with the ethyl acetate fractions from the leaves of *K. tomentosa* has exhibited antidiabetic properties by means of the inhibition of the enzyme α-amylase [29].

This work aimed to obtain extracts of distinct polarity from the leaves of *K. tomentosa*, and evaluate their phytochemical composition by chromatography techniques such as GC/MS and UPLC/MS/MS. In addition, it sought to demonstrate the antibacterial activity of the obtained extracts against gram-positive and gram-negative bacteria, as well as their antioxidant activity towards DPPH, ABTS, and H_2_O_2_ radicals. This study also integrated *in vivo* models to evaluate their possible toxicity effects. *In silico* modeling was implemented to propose a possible antibacterial mechanism of the compounds contained in the methanol extract against gram-positive strains. The obtained results expand the knowledge about the potential therapeutic use of *K. tomentosa* and assess the importance of continuing to study the biological activities of species from the genus *Kalanchoe*.

## 2. Results

### 2.1. Total Flavonoid Content (TFC) and FTIR Analyses

The TFC of the methanol extract from *K. tomentosa* was assessed by constructing a calibration curve constructed with quercetin at 100, 150, 200, 250, and 300 µg/mL (see Figure 1). Considering the obtained data from this figure, the TFC of the methanol extract of *K. tomentosa* was 72.46 mg Qu/g.

### 2.2. GC/MS Analysis

The GC/MS analyses of the hexane and chloroform extracts from *K. tomentosa* are presented in Table 1. Appendix A represent the obtained chromatograms.

### 2.3. UPLC/MS/MS Analysis

As depicted in Figure 2, the phytoconstituents of the methanol extract from *K. tomentosa* are kaempferol-3-*O*-rutinoside (**2**; Rt: 7.6 min), quercetin-*O*-hexoside (**3**; Rt: 8.8 min), kaempferide-3-glucuronide (**4**; Rt: 8.9 min), eriodictyol-7-*O*-hexoside (**5**; Rt: 9.0 min), deacetoxy (7)-7-oxokhivorinic acid (**6**; Rt: 9.2 min), kaempferol 3-*O*-α-L-arabinopyranosyl-(1→2) α-L-rhamnopyranoside (**7**; Rt: 10.0 min), kaempferin (**8**; Rt: 10.2 min), ganolucidic acid C (**9**; Rt: 10.7 min), kaempferol (**10**; Rt: 11.4 min), spiraeoside (**11**; Rt: 11.5 min), apigenin (**12**; Rt: 12.4 min), linoleic acid (**13**; Rt: 19.8 min), and heliannuol A (**14**; Rt: 20.7 min). Despite these findings, compound **1** (Rt: 0.8 min) was not identified. Appendix A provides the retention times of the identified compounds in this extract and the references used to compare them.

### 2.4. Antibacterial Activity

As illustrated in Figure 3A, it can be observed that treatment with 50, 100, and 150 μg/mL of hexane extract caused the death of 20.43 ± 3.85, 28.23 ± 9.40, and 38.03 ± 9.57% *E. coli* cells, respectively. At 200, 250, and 300 μg/mL, treatment with the same extract occurred in 42.5 ± 3.49, 42.76 ± 3.01, and 53.16 ± 1.20% death cells. In comparison, treatment with 50 and 100 μg/mL of chloroform extract induced the death of 24.56 ± 4.05 and 28.93 ± 0.83% *E. coli* cells, whereas at 150, 200, and 250 μg/mL, treatment resulted in 34.40 ± 3.67, 34.60 ± 2.57, and 35.56 ± 2.58% death cells, respectively. The highest activity of chloroform extract against *E. coli* was determined at 300 μg/mL, as it resulted in 35.86 ± 3.04% death cells. Contrarily to these results, treatment with methanol extract at 50, 100, and 150 μg/mL resulted in the death of 53.60 ± 3.10, 55.36 ± 1.00, and 59.46 ± 1.16% cells. At 200, 250, and 300 μg/mL, methanol extract caused 59.46 ± 1.16, 63.16 ± 3.63, and 63.73 ± 1.94% death cells. Fosfomycin was considered a positive control against *E. coli*, and it was tested at 300 μg/mL, resulting in 86.50 ± 1.50% death cells.

The activity of the hexane, chloroform, and methanol extracts from *K. tomentosa* towards *K. pneumoniae* is depicted in Figure 3B. As seen, hexane extract at 50, 100, and 150 μg/mL resulted in 34.00 ± 3.40, 42.53 ± 0.80, and 42.70 ± 0.70% death cells, respectively. At 200, 250, and 300 μg/mL, treatment with the same extract occurred in 43.06 ± 1.18, 43.36 ± 1.51, and 42.70 ± 0.79% of *K. pneumoniae* death cells. Similarly, treatment with 50, 100, and 150 μg/mL of chloroform extract caused the death of 25.66 ± 1.87, 28.63 ± 4.56, and 35.16 ± 4.05% cells, respectively. At 200, 250, and 300 μg/mL, treatment with chloroform extract resulted in 42.90 ± 2.17 and 48.80 ± 1.13% *K. pneumoniae* cells. On the other hand, methanol extract at 50 and 100 μg/mL caused 25.23 ± 2.48 and 34.90 ± 1.50% cell death. At 150, 200, and 250 μg/mL, it promoted the death of 42.10 ± 0.88, 45.80 ± 2.6, and 63.46 ± 1.09% of cells. The highest activity of methanol extract against *K. pneumoniae* was recorded at 300 μg/mL since it caused the death of 68.40 ± 1.34% of cells. These findings were compared to treatment with 300 μg/mL of amikacin, which resulted in 88.06 ± 3.30% *K. pneumoniae* death cells.

The capacity of extracts from *K. tomentosa* to decrease the viability of *S. aureus* is represented in Figure 3C. As presented, 50 and 100 μg/mL of hexane extract resulted in 21.83 ± 0.60 and 25.53 ± 1.36% death cells, whereas treatment at 150 and 200 μg/mL occurred in 25.63 ± 5.15 and 26.70 ± 0.98 *S. aureus* death cells. For this extract, the highest activity was established during treatment at 250 and 300 μg/mL, which resulted in 27.00 ± 1.38 and 27.26 ± 6.18 death cells, respectively. Comparably, treatment with 50, 100, and 150 μg/mL of chloroform extract caused the death of 45.43 ± 0.30, 46.20 ± 1.24, and 51.80 ± 0.20% cells. At 200, 250, and 300 μg/mL, chloroform extract occurred in 56.56 ± 2.02, 62.66 ± 5.92, and 65.53 ± 3.45% *S. aureus* death cells. Among the obtained extracts from *K. tomentosa*, methanol extract exerted the highest activity towards *S. aureus* since it resulted in the death of 70.73 ± 0.92, 72.36 ± 0.58, and 74.20 ± 0.36% death cells at 50, 100, and 150 μg/mL, respectively. In comparison, treatment at 200, 250, and 300 μg/mL promoted the death of 74.53 ± 0.47, 76.86 ± 0.55, and 79.30 ± 0.62% of *S. aureus* death cells. Vancomycin was appraised as a positive control, causing the death of 85.56 ± 4.05 cells at 300 μg/mL. Table 2 compiles the MIC_90_ values of extracts against each strain.

### 2.5. Analysis of Antibacterial Activity In Silico

The interaction between compounds **2**, **7**, **9**, and **11** with the NAM/NAG-peptide subunits is schematized in Figure 4. The binding energies resulting from these interactions are presented in Table 3. Appendix A can be consulted to observe the molecular docking studies (Appendix A). and binding energies (Appendix A) of the rest of the identified compounds by UPLC/MS/MS, and vancomycin.

### 2.6. Antioxidant Activity

#### 2.6.1. DPPH Assay

As noted in Figure 5A, it can be noted that 50 and 100 µg/mL of hexane extract occurred in the scavenging of 55.53 ± 0.03 and 55.75 ± 0.23% DPPH radicals, respectively. In contrast, treatment with 150, 200, and 250 µg/mL inhibited the generation of 56.08 ± 0.74, 56.63 ± 0.36, and 57.06 ± 0.11% DPPH radicals. For this extract, the highest percentage of inhibited DPPH radicals was recorded at 300 µg/mL, which resulted in 57.72 ± 0.18% inhibited DPPH radicals. Moreover, it can be observed that treatment with 50 µg/mL of chloroform extract scavenged 35.76 ± 1.02% of DPPH radicals, whereas treatment with 100 and 150 µg/mL inhibited the generation of 36.07 ± 0.73 and 36.07 ± 0.75% of DPPH radicals, respectively. Comparably, 36.17 ± 1.93% DPPH radicals were scavenged at 200 and 250 µg/mL of the same extract. The highest antioxidant activity of chloroform extract was achieved at 300 ± 0.89 µg/mL, as it resulted in the scavenging of 36.48% of DPPH radicals. The antioxidant activity of *K. tomentosa* was different for the methanol extract since it can be noted that treatment with 50, 100, and 150 µg/mL inhibited the generation of 91.04 ± 0.14, 91.25 ± 0.14, and 91.68 ± 0.12% of DPPH radicals, respectively. The same extract at 200, 250, and 300 µg/mL scavenged 91.78 ± 0.07, 91.89 ± 0.07, and 92.21 ± 0.50% of DPPH radicals, respectively. Ascorbic acid (AC) was used to compare the antioxidant activity of these extracts. As presented in the same figure, 50 µg/mL of AC scavenged 90.94 ± 0.26% DPPH radicals. Even though no differences were observed at 100 µg/mL, it can be noted that treatment with 150, 200, and 250 µg/mL inhibited the generation of 91.05 ± 0.52, 91.16 ± 0.29, and 91.16 ± 1.63% DPPH radicals, respectively. At 300 µg/mL, treatment with AC resulted in the scavenging of 92.21 ± 1.16% of radicals.

#### 2.6.2. ABTS Assay

As illustrated in Figure 5B, treatment with 50 and 100 µg/mL of hexane extract inhibited 32.54 ± 1.05 and 32.63 ± 1.33% ABTS radicals, respectively. Moreover, at 150, 200, and 250 µg/mL, hexane extract scavenged 34.21 ± 1.61, 34.47 ± 2.11, and 34.95 ± 0.07% free radicals, respectively. At 300 µg/mL, treatment resulted in the inhibition of 35.43 ± 2.31% radicals. Comparably, treatment with 50, 100, and 150 µg/mL of chloroform extract resulted in the scavenging of 36.09 ± 4.18, 36.44 ± 4.20, and 36.97 ± 3.50% ABTS radicals, respectively. At 200, 250, and 300 µg/mL, treatment with extract inhibited the formation of 37.19 ± 1.24, 37.76 ± 1.91, and 38.50 ± 0.77% radicals. In comparison to hexane and chloroform extracts, treatment with 50, 100, and 150 µg/mL of methanol extract inhibited the formation of 34.21 ± 4.67, 36.84 ± 3.08, and 36.97 ± 0.22% free radicals, respectively. At 200, 250, and 300 µg/mL, treatment with methanol extract scavenged 39.78 ± 0.10, 53.55 ± 2.55, and 70.43 ± 9.51% ABTS radicals. The activity of extracts against ABTS radicals was compared to AC. As represented in Figure 5B, treatment with 50, 100, and 150 µg/mL of AC scavenged 56.62 ± 3.38, 57.41 ± 3.08, and 63.28 ± 0.22% free radicals, respectively. At 200 and 250 µg/mL, treatment with AC inhibited 66.01 ± 0.10 and 70.30 ± 2.55% radicals, whereas at 300 µg/mL it inhibited the formation of 83.33 ± 9.51% ABTS radicals, respectively. No significant differences were recorded in comparison to the positive control.

#### 2.6.3. H_2_O_2_ Assay

As seen in Figure 5C, treatment with 50, 100, and 150 µg/mL of hexane extract scavenged 23.49 ± 0.03, 27.81 ± 0.03, and 29.89 ± 0.10% H_2_O_2_ radicals, respectively. At 200, 250, and 300 µg/mL, treatment with extract inhibited the generation of 33.89 ± 0.01, 41.19 ± 0.09, and 43.61 ± 0.02% radicals. Contrarily, treatment with chloroform extract at 50 and 100 µg/mL inhibited 47.33 ± 0.59 and 53.27 ± 0.09% free radicals, respectively. At 150, 200, and 250 µg/mL, treatment with chloroform extract scavenged 54.25 ± 0.02, 58.63 ± 0.04, and 68.38 ± 0.07% radicals, respectively. For chloroform extract, the highest activity against H_2_O_2_ was recorded at 300 µg/mL since it inhibited the formation of 76.35 ± 0.04% radicals. Moreover, treatment with 50 and 100 µg/mL of methanol extract inhibited 45.11 ± 0.09 and 47.25 ± 0.07% free radicals, whereas treatment at 150, 200, and 250 µg/mL scavenged 52.78 ± 0.02, 54.80 ± 0.07, and 86.99 ± 0.12% H_2_O_2_ rathe IC50dicals. At the highest concentration proposed in this work, treatment with 300 µg/mL resulted in the inhibition of 98.27 ± 0.05% radicals. The antioxidant activity of extracts from K. tomentosa against ABTS radicals was compared to AC. As observed in the same figure, treatment with 50 and 100 µg/mL of AC inhibited 81.26 ± 0.03 and 83.15 ± 0.06% radicals, whereas at 150 and 200 µg/mL, treatment scavenged 83.23± 0.08 and 84.81 ± 0.05% ABTS radicals, respectively. At 250 and 300 µg/mL, treatment with AC inhibited the formation of 90.49 ± 0.05 and 93.57 ± 0.05% free radicals. No significant differences were determined against the positive control. Considering the obtained results from the DPPH, ABTS, and H_2_O_2_ assays, the half-maximal inhibitory concentration (IC_50_) values for each extract were calculated and compiled in Table 4.

### 2.7. Evaluation of In Vivo Toxicity in A. salina and C. elegans

As represented in Figure 6A, treatment with hexane, chloroform, and methanol extracts from K. tomentosa at the proposed concentrations (50, 100, 150, 200, 250, and 300 μg/mL) did not decrease the viability of *A. salina* nauplii, and it did not cause any morphological changes after 24 h exposure to treatment. In comparison, treatment with K_2_Cr_2_O_7_, a frequently used positive control in toxicity assays, resulted in a 10% survival rate (90% nauplii death). Contrary to these results, treatment with extracts exerted high toxicity against *C. elegans* nematodes.

As seen in Figure 6B, hexane extract at 50 μg/mL resulted in a 93.33–17.78% *C. elegans* survival rate during exposure to treatment for 6 h. During the same period, treatment with 100 μg/mL of hexane extract resulted in an 86.67–2.22% survival rate, whereas treatment at 150 and 200 μg/mL caused a 73.33–1.00% survival rate; this effect endured after treatment with 250 and 300 μg/mL of hexane extract, respectively. In the case of chloroform extract, it was determined that treatment at 50 μg/mL caused a 77.78–17.78% survival rate after 4 h of exposure to treatment, which rapidly reduced to 1.00% after 5 and 6 h. For this extract, the highest toxicity was recorded at 150–300 μg/mL since it caused a 73.33–2.22% survival rate (see Figure 6C). Among the obtained extracts, methanol extract performed the highest toxicity activity against *C. elegans* nematodes, as it caused 26.67 and 6.67% survival rates at 50 and 100 μg/mL, respectively. Moreover, it was only possible to assess the resultant survival rates at 0 h due to the high toxicity of the extract. In this context, it was observed that treatment with 150, 200, 250, and 300 μg/mL occurred in 11.11, 6.67, 4.44, and 1.00% survival rates (see Figure 6D).

## 3. Discussion

The genus *Kalanchoe* includes a wide variety of species utilized to prepare juices, salads, infusions, or extracts for medicinal purposes. The biological activities of *Kalanchoe* species are attributed, predominantly, to the presence of flavonoids and polyphenols, since only a limited number of studies have investigated their nonpolar counterparts. Here, the leaves of *K. tomentosa* were utilized to obtain nonpolar (hexane and chloroform) and polar (methanol) extracts by maceration, which were consequently evaluated by spectroscopy methods, chromatography approaches, and *in vitro*, *in vivo*, and *in silico* techniques.

Among analytical techniques, FTIR spectroscopy is characterized by its importance in assessing the chemical composition of organic and inorganic analytes. In comparison to other spectroscopy approaches, this event arises from the capacity of molecules to absorb infrared radiation and generate distinct molecular vibrations, such as stretching, bending, rocking, and wagging vibrations. In accordance with Figure 1A, the hexane, chloroform, and methanol extracts from *K. tomentosa* presented a series of similar peaks within the range of 4000–500 cm^−1^. Bands localized at 3200 cm^−1^ can be associated with secondary metabolites containing hydroxyl groups within their structure. Moreover, it can be noted that the three extracts displayed two small peaks at 2970–2850 cm^−1^, which can be related to the asymmetric and symmetric stretching of methyl groups. In contrast to hexane extract, chloroform and methanol extracts exhibited wide and small bands located in 1750–1735 cm^−1^, which can correspond to carbonyl, ketones, or esters groups from the presented identified compounds. The rest of the peaks recorded from 1500 to 1000 cm^−1^ can be correlated to the bending of carbon and hydrogen bonds possible in molecules constituted by aromatic rings. The decrease in transmission below 1000 cm^−1^ can be related to the stretching of carbon and oxygen bonds or the deformation of oxygen and hydrogen bonds possibly contained among the identified compounds: amyrin, lupenone, or the derivatives of quercetin and kaempferol. Within the same range, peaks can be associated with the presence of inorganic matter, such as minerals or salts, probably contained in the raw material and solubilized during the extraction process. The obtained findings during FTIR analyses enable us to establish the chemical fingerprint of the hexane, chloroform, and methanol extracts from *K. tomentosa*.

There are different techniques to estimate the TFC of extracts from medicinal plants, and its preferably implemented to analyze polar extracts since they possess compounds characterized by their numerous polar functional groups that can react with specific reagents to produce colored complexes, which are not obtained from nonpolar extracts such as hexane and chloroform extracts. One of these reagents includes the use of AlCl_3_, a white and crystalline solid that, due to its coordination complexing capacity, can interact with the hydroxyl and keto groups of flavonoids and, herein, allow their quantification utilizing UV-vis spectrophotometry. Flavonoids that can be used as standards to perform the TFC assay comprehend quercetin (Qu), rutin (Ru), and catechin, and they are necessary for the calibration and quantification process. Conforming to Figure 1B, a calibration curve was constructed with Qu at 100, 150, 200, 250, and 300 μg/mL and used to obtain the following regression equation: *y* = 0.0028*x* + 0.0727, R^2^ = 0.993. Here, *y* was appraised as the absorbance of the test sample, and *x* was contemplated as the concentration obtained from the calibration curve. According to this, the TFC of the methanol extract of *K. tomentosa* was 72.46 mg Qu/g, which is similar to the hydroethanolic extract from *K. brasiliensis* (16.95 mg Ru/g) [30], but lower than *K. fedtschenkoi* (384.54 mg Qu/g) [31]. The discrepancies between other reported TFC values and the obtained results in this work can be attributed to the method used to prepare extracts, solvent, amount of raw material, and geographic location. After preliminary phytochemical evaluation, the chemical composition of extracts from *K. tomentosa* was studied through GC/MS and UPLC/MS/MS.

Chromatography techniques are required to separate and identify diverse components from complex organic and inorganic mixtures. In contrast to other chromatography approaches, GC/MS can be used to analyze biological samples and volatile compounds from plant extracts. In this sense, nonpolar extracts are frequently analyzed by GC/MS since they contain compounds that tend to exhibit higher volatility compared to polar compounds, stronger interactions with GC column stationary phases, and compatibility with mass spectrometric detectors [32]. The GC/MS analyses of the hexane and chloroform extracts from *K. tomentosa* are compiled in Table 1 and indicate that the hexane extract is constituted by phytosterols, such as β-sitosterol, and various triterpenoids, such as α- and β-amyrin, α-amyrone, lupen-3-one, lupenone, and urs-12-ene. The presence of the same compounds was recorded in the chloroform extract, together with multiple fatty acids such as heptacosanol and the acetate derivative of β-amyrin. The presence of these compounds can be associated with data retrieved from FTIR analyses; however, fatty acids were not determined in the hexane extract since only compounds with a high R match factor (≥800) were considered during GC/MS evaluation.

In medicinal plants, phytosterols constitute a broad category of natural products derived from steroids that can exhibit antimicrobial, antioxidant, anti-inflammatory, antidiabetic, and neuroactive activities [33]. In *K. tomentosa*, recent studies reported the isolation of β-sitosterol through vacuum liquid chromatography and its recrystallization using *n*-hexane [27]. The presence of β-sitosterol in a methanolic fraction obtained from the leaves of *K. pinnata* has also been demonstrated [34]. Triterpenoids are secondary metabolites derived from isopentenyl pyrophosphate oligomers that are recognized because of their antiallergic, antiviral, antiangiogenic, and spasmolytic properties [35]. Pentacyclic triterpenoids, such as α- and β-amyrin, are limitedly produced by plants but execute strong anxiolytic, antidepressant, anticonvulsant, gastroprotective, and hepatoprotective activities *in vivo* [36,37]. In species from the genus *Kalanchoe*, α- and β-amyrin have been identified in *K. pinnata* and *K. daigremontiana* [19,38], whereas their acetate derivatives have been crystallized in *K. pumila* [39]. However, the scientific evidence regarding the existence of triterpenoids such as lupeol, lupenone, or lupen-3-one in extracts from *Kalanchoe* species has not been reported yet. The GC/MS identification of lupeol and lupen-3-one in the hexane and chloroform extracts from *K. tomentosa* is of great importance since triterpenoids such as lupeol can exert strong antihyperglycemic, antimutagenic, and antioxidant properties in *in vitro* and *in vivo* models [40,41,42], whereas lupane-type triterpenoids like lupen-3-one are attractive natural products for drug development as they can display antitumor, anti-inflammatory, and antidiabetic activities [43].

UPLC/MS/MS is a reliable, time- and cost-effective approach that possesses increased resolution, selectivity and sensibility, compatibility with various ionization techniques, and yields essential chemical information about organic components. In the study of phytochemicals with promising applications in biomedicine and drug development, UPLC-MS/MS analyses are useful to standardize, authenticate, and identify the constituents of many substances. UPLC/MS/MS analysis is preferred to analyze polar extracts since it enables better separation of polar compounds from complex matrices by providing optimal experimental conditions to detect and quantify them [44]. In addition, it is highly implemented since polar compounds tend to be thermally labile, which cannot be easily controlled during GC/MS analyses. In this regard, UPLC/MS/MS analysis of the methanol extract of *K. tomentosa* yielded fourteen compounds, of which only one was not identified when compared to other studies where similar techniques have been applied to study the phytoconstituents of polar extracts from *Kalanchoe* species, such as *K. pinnata* [45,46], *K. brasiliensis* [47], *K. laxiflora* [48], and *K. gastonis-bonnieri* [49]. In this context, the identified compounds in methanol extract were kaempferol-3-*O*-rutinoside (**2**), quercetin-*O*-hexoside (**3**), kaempferide-3-glucuronide (**4**), eriodic-tyol-7-*O*-hexoside (**5**), deacetoxy (7)-7-oxokhivorinic acid (**6**), kaempferol 3-*O*-α-L-arabinopyranosyl-(1→2) α-L-rhamnopyranoside (**7**), kaempferin (**8**), ganolucidic acid C (**9**), kaempferol (**10**), spiraeoside (**11**), apigenin (**12**), linoleic acid (**13**), and heliannuol A (**14**). The determination of glycosidic derivatives of flavonoids is in accordance with the fact that polar extracts from *Kalanchoe* species are abundant sources of these compounds and establishes the possibility that methanol extracts from *K. tomentosa* contain novel natural products with other potential biological activities, such as antitumoral, antihypertensive, and antidiabetic [50,51,52].

Over the last decades, bioactive compounds from medicinal plants have attracted special attention to the design of novel strategies to mitigate challenges related to the treatment of infections caused by multidrug-resistant bacteria. In health care settings, the prevalence of high priority strains such as *E. coli* has resulted in numerous cases of urinary tract, bloodstream, or wound infections [53]. Similarly, *K. pneumoniae*, a gram-negative critical priority pathogen, can also cause these infections together with liver abscess, or pneumonia [54,55]. In contrast to gram-negative strains, gram-positive bacteria such as *S. aureus* are associated with respiratory, bone and joint, and catheter-associated infections [56,57]. Here, it was assessed that the hexane, chloroform, and methanol extracts from *K. tomentosa* performed poor activity against *E. coli*, *K. pneumoniae*, and *S. aureus* by means of their MIC_90_ values (see Table 2). The obtained results suggest that the cultured strains are resistant to treatment since their growth was not significantly inhibited during treatment with extracts after 24 h below the range of 50–300 μg/mL. Interestingly, it was noted that treatment with 300 μg/mL of methanol extract exerted similar effects as vancomycin at the same concentration. Therefore, it was decided to continue exploring this event in *in silico* studies.

Current research fields aim to implement *in silico* screening to evaluate the binding interactions between secondary metabolites from medicinal plants with target protein moieties or receptors of therapeutic interest. Vancomycin, a glycopeptide antibiotic, is prescribed to treat infections caused by gram-positive bacteria since it can inhibit the biosynthesis of their cell-wall by forming hydrogen bonds with D-alanyl-D-alanine moieties from the NAM/NAG-peptide and hence, hamper its incorporation into the peptidoglycan matrix. According to the thirteen compounds identified by UPLC/MS/MS analysis, *in silico* modeling was applied to evaluate their binding with the aminoacidic moieties from the NAM/NAG-peptide (PDB ID: 2EAX) (see Appendix A). Based on the calculated binding energies, only compounds **2** (−6.1 ± 0.4 Kcal/mol), **7** (−5.9 ± 0.3 Kcal/mol), **9** (−6.3 ± 0.5 Kcal/mol), and **11** (−5.7 ± 0.4 Kcal/mol) were demonstrated to interact with the NAM/NAG-peptide by forming hydrogen bonds with the LYS 504, DAL 505, and DAL 506 residues. In contrast to the binding energy of vancomycin (−3.1 ± 0.7 Kcal/mol), it was revealed that the identified products in the methanol extract from *K. tomentosa* exhibit higher interaction with the NAM/NAG-peptide. These results can explain the recorded activity of methanol extract against *S. aureus* and suggest a possible novel antibacterial mechanism of quercetin, kaempferol, and apigenin, or their derivatives. The obtained results are demanding to compare since there are no studies where the antibacterial activity of extracts from *Kalanchoe* species has been studied by similar approaches. However, it has been documented that they have been considered to analyze the capacity of bufadienolides (e.g., bryotoxin, hovetrichoside C, and bersaldegenin 1-acetate) from *K. daigremontiana* to inhibit plasmin, a serine protease enzyme with major functions in clot formation and dissolution [58]. In the same regard, it has been reported that constituents of *K. pinnata,* such as palmitic acid, can exert strong anti-inflammatory properties by means of their interaction with ligands, enzymes, and nuclear receptors involved in inflammatory phenomena [59]. Using extracts with similar polarity, *in silico* approaches have been applied to evaluate the pharmacokinetic features and antidiabetic potential of *K. daigremontiana* [60], and *K. pinnata* [61].

Medicinal plants can exert antioxidant properties due to the capacity of their phytochemical content to scavenge the generation of free radicals [62], and modulate the activity of antioxidant enzymes such as superoxide dismutase and catalase [63]. The disruption of the formation of free radicals by extracts from medicinal plants is significant since it suggests their possible application to mitigate the initiation or progression of oxidative-stress-related disorders such as Parkinson’s disease [64], cancer [65], or diabetes [66]. Here it was noted that treatment with hexane extract exerted the highest scavenging capacity towards DPPH radicals than chloroform extract (*p* < 0.001). Contrarily, methanol extract executed significant antioxidant activity against the same radical in comparison to the hexane and chloroform extracts, and AC (*p* < 0.001). Again, statistical analysis revealed that methanol extract performed the highest antioxidant activity of ABTS radicals in a dose-dependent manner (50–300 μg/mL). During their evaluation of H_2_O_2_ radicals, it was recorded that chloroform extract can significantly inhibit their formation (*p* < 0.001), whereas methanol extract can exert the highest antioxidant activity than AC (*p* < 0.001). The obtained results can be attributed to the synergy between the different secondary metabolites contained in each extract.

According to their IC_50_ values, it was revealed that the hexane, chloroform, and methanol extracts from *K. tomentosa* can exert high or moderate antioxidant activity, depending on the implemented assay. Against DPPH radicals, the antioxidant performance of the hexane (IC_50_ 2.76 μg/mL) and methanol (IC_50_ 3.95 μg/mL) extracts can be appraised as high, whereas the activity of the chloroform extract can be considered low (IC_50_ 6209.56 μg/mL). Contrarily, the antioxidant activity of the hexane, chloroform, and methanol extracts towards ABTS radicals can be associated with a moderate effect due to their IC_50_ values: 978.14, 1554.73, and 210.22 μg/mL, respectively. Again, the antioxidant capacity of the hexane (IC_50_ 376.83 μg/mL), chloroform (IC_50_ 87.81 μg/mL), and methanol (IC_50_ 110.75 μg/mL) extracts during the H_2_O_2_ assay can be related to a moderate effect. The variabilities between the antioxidant performance of extracts can be attributed to differences in the sensibility and reaction kinetics of each selected assay, which have been recently observed by our research group and supported with additional approaches, such as machine learning modeling [67]. In addition, the antioxidant activity of extracts from *K. tomentosa* was enhanced in accordance with their polarity rather than a dose-dependent effect, making the methanol extract the most antioxidant one.

The antioxidant activity of extracts of species from the genus *Kalanchoe* has been mainly studied for their polar extracts but barely reported for their nonpolar extracts. Recent scientific evidence demonstrated that fractions of ethanol extracts from *K. daigremontiana*, *K. pinnata*, *K. milloti*, and *K. nyikae* can inhibit the generation of DPPH radicals at IC_50_ values of 180, 90.6, 61.5, and 341 µg/mL, respectively [68]. Comparably, water extracts from *K. daigremontiana* have been documented to scavenge DPPH radicals at IC_50_ 1750 µg/mL [69]. Following the ABTS or H_2_O_2_ assays, a water fraction from *K. blossfeldiana* has been proven to inhibit the generation of radicals at IC_50_ 3.17 μg/mL [21], whereas other studies reported that fractions from the extract from *K. pinnata* can scavenge ABTS and H_2_O_2_ radicals at different IC_50_ values [70]. The antioxidant activity of *Kalanchoe* species is correlated with the abundant presence of flavonoids such as quercetin and kaempferol, or bufadienolides such as bryophyllin and bersaldegenin. In the case of the hexane and chloroform extracts from *K. tomentosa*, their antioxidant performance is challenging to compare since the evidence regarding their evaluation to quench DPPH, ABTS, or H_2_O_2_ radicals is scarce. For compounds contained in the methanol extract, it has been reported that quercetin and kaempferol can scavenge DPPH, ABTS, and H_2_O_2_ radicals by their hydrogen- and electron-donating capabilities, which occur in neutralized free radicals. Similar effects can be expected from the rest of the compounds identified through UPLC/MS/MS analyses since they possess multiple hydroxyl groups distributed within their chemical structures.

Toxicity assays are necessary to establish the potential hazards of a variety of substances, such as plant extracts. They are executed among living organisms, such as *A. salina* nauplii and *C. elegans* nematodes. The former belongs to the genus *Artemia* and constitutes a group of small, transparent, and elongated aquatic crustaceans with wide importance in aquaculture as a food source and biomedical research to evaluate the toxicity of organic and inorganic substances [71]. The latter are transparent, nonparasitic worms with simple anatomy and a short life cycle that corresponds to the genus *Caenorhabditis* useful to investigate neurobiological behavior and gene expression [72], and potential toxicity effects [73]. According to Figure 6, treatment with extracts did not compromise the viability of *A. salina* nauplii nor their anatomical features after exposure to treatment with 50–300 μg/mL of the hexane, chloroform, and methanol extracts. Even though these results suggest the biocompatibility of extracts, it is worthy to observe that extracts were highly toxic to *C. elegans* nematodes after 6 h of exposure to treatment. We hypothesize that discrepancies between the collected data can be associated with sensitivity, metabolic capacity, and detoxification differences between *A. salina* and *C. elegans*. Further studies are required to investigate such events since current studies about the biological activities of medicinal plants are primarily focused on determining their effect on survival rates. On the other hand, it is relevant to highlight that this is the first study where *C. elegans* nematodes are cultured to analyze the *in vivo* toxicity of extracts from *Kalanchoe* species.

## 4. Materials and Methods

### 4.1. Plant Collection and Identification

Leaves from *K. tomentosa* were collected in Tlaxcalancingo, San Andrés Cholula, Puebla (19°0′36″ N, 98°15′36″ W). The biologist Lucio Caamaño Onofre identified the specimens, and they were deposited in Jardín Botánico Universitario of the Benemérita Universidad Autónoma de Puebla (BUAP; 24 Sur Av. San Claudio, Col. San Manuel C.P. 72570 Puebla de Zaragoza, Puebla, México) with the voucher number 89257.

### 4.2. Extracts Preparation

Initially, the leaves from *K. tomentosa* were washed with distilled water and air-dried at room temperature for two weeks. After this, leaves were powdered using a mechanical blender and then, macerated consecutively with 2 L of hexane, chloroform, and methanol for three days, respectively. Mixture was filtrated, and solvent was removed by evaporation to dryness under reduced pressure utilizing a Heidolph Laborota 4000 efficient rotary evaporator (Heidolph, Schwabach, Germany). Extracts were maintained under refrigeration until further evaluation.

### 4.3. TFC Analysis 

The TFC from the methanol extract from *K. tomentosa* was determined by incubating 100 µL of each extract with 100 µL of 2% aluminum chloride for 10 min. Then, the sample was placed in 1 cm quartz cuvettes, and absorbance was recorded using a Cary 60 UV-Vis spectrophotometer at 420 nm (Agilent Technologies, Santa Clara, CA, USA). Since a standard curve was developed utilizing Qu at various concentrations, the TFC of the extract was reported as a percentage of total Qu equivalents per gram of extract (mg Qu/g). Experiments were performed by triplicate.

### 4.4. FTIR Analysis

A Cary 630 Fourier-transform infrared (FTIR) spectrometer (Agilent Technologies, Santa Clara, CA, USA) was used to analyze the general composition of extracts from *K. tomentosa*. Briefly, the detection diamond was cleaned with an ethanol solution (100% *v*/*v*) and enabled to air dry. Subsequently, 20 mg of each extract were placed onto the detection diamond, and measurements were determined within the 4000 to 400 cm^−1^ regions. Experiments were conducted in triplicate.

### 4.5. GC/MS Analysis

The chemical composition of the hexane and chloroform extracts from *K. tomentosa* was evaluated using an Agilent Technologies 6850 Network GC System (Santa Clara, CA, USA) coupled to an Agilent 5975C VL Mass Selective Detector (MSD). Briefly, 10 µL of each sample (1 mg/mL) were injected into a HP-5MS (5% Phenyl Methyl Siloxane; 30.0 m × 250 µm × 0.25 µm nominal) column from Agilent Technologies, and the following gradient temperature was set: 60 °C for 5 min, 80 °C/min up to 200 °C for 10 min, and 10 °C/min up to 240 °C for 15 min. For each experiment, helium was used as the carrier gas at a flow rate of 11.3 mL/min. The individual components were assessed in accordance with their retention times, match factor, and reverse match factor (R match), which are reported in the National Institute of Standards and Technology Mass Spectral (NIST-MS) database. Only compounds with a high R match (≥800) were reported in Section 2.

### 4.6. UPLC/MS/MS Analysis

The phytochemical profile of methanol extract from *K. tomentosa* was determined utilizing an Ultimate 3000 UPLC system (Dionexcorp, Sunnyvale, CA, USA) coupled with a Bruker Micro-TOF-QII mass spectrometer equipped with an electrospray ionization (ESI) source (Bruker Daltonics, Billerica, MA, USA) operating in positive ionization mode. The Data Analysis 4.0 software (Bruker Daltonics, Bremen, Germany, Technical Note 008, 2004) was utilized to process the acquired fragmentation data of compounds. A Varian Hypersil C18 chromatography column (3.0 µm, 125 × 4.0 mm) was selected to perform the analysis. Mobile phase consisted of 0.1% formic acid (A) and acetonitrile (B), and gradient temperature consisted of 0–10 min (5–35% B), 10–21.1 min (35–80%), and 21.1–30 min (80–80%). Flow rate was 0.5 mL/min, column temperature was adjusted to 30 °C, and absorbance was recorded at 254 nm. For mass spectrometry analysis, capillary voltage was 4500 V, drying gas temperature was 180 °C, gas flow rate was 8 L/min, and 2.5 bar ESI nebulizer gas pressure. The mass range of measurements was 50–3000 *m*/*z.*

### 4.7. Strains, Culture Media, and Antibacterial Assay

The antibacterial activity of extracts from *K. tomentosa* was evaluated against gram-positive and gram-negative strains. Gram-negative bacteria strains comprised *E. coli* (ATCC 25922) and *K. pneumoniae* (ATCC 14210). Gram-positive bacteria were *Staphylococcus aureus* (ATCC 25923). All strains were cultured in Mueller–Hinton broth (B&D) at 37 °C using a Julabo GmbH SW22 thermostatic water bath (Julabo GmbH, Seelbach, Germany). The microdilution assay was executed to determine the effect of the extracts from *K. tomentosa* on the viability of bacteria strains. Shortly, 100 μL of Mueller-Hinton broth were dispensed (per well) in a flat-bottomed 96-well plate together with bacteria inoculum prepared to have a final optical density of 0.05 at 600 nm, and extracts at 50, 100, 150, 200, 250, and 300 μg/mL. Plate was maintained at 37 °C overnight under orbital shaking, and the absorbance of wells was recorded at 600 nm in a Multiskan Sky microplate reader (Thermo Scientific, Waltham, MA, USA). Fosfomycin, amikacin, and vancomycin were considered positive controls against *E. coli*, *K. pneumoniae*, and *S. aureus*, respectively. All experiments were performed in triplicate.

### 4.8. In Silico Analysis

The interaction between compounds from the methanol extract from *K. tomentosa* with the NAM/NAG peptide of the peptidoglycan matrix from *S. aureus* was investigated through *in silico* modeling. Briefly, AutodockFR program v.1.0 was selected for docking test, and modules were installed and ran in Ubuntu 22.04.4 LTS operative system with an Intel^®^ Core™ i7-13700K and 32 GB RAM. Peptidoglycan pentapeptide was retrieved from PDB ID:2EAX, and Obabel module was used to convert pentapeptide pdb file to pdbqt. Flavonoids structures were designed in MarvinSketch 23.11. Structures optimizations were carried out with Gaussian16 (1) at PM6 semiempiric quantum mechanics level of theory. The module prepare_ligand.py from ADFR was used to convert pdb files from optimization to pdbqt required for docking. AGFRGUI module was employed to generate target file maps and search boxes all around the entire peptidoglycan. One hundred runs of Lamarckian genetic algorithm were performed for each ligand at 25,000,000 maximums of searching evaluations. Results were retrieved from best cluster rank energy. Three-dimensional interaction figures were performed on ChimeraX v.1.6.

### 4.9. Antioxidant Assays

The antioxidant activity of the extracts from *K. tomentosa* was tested against DPPH, ABTS, and H_2_O_2_ radicals. For the DPPH assay, a 1 mM solution was prepared by dissolving 4 mg of DPPH reagent in technical grade ethanol, which was maintained under moderate stirring for 2 h in a dark place. Consequently, 200 μL of the resultant solution were placed, per well, in a 96-well plate and mixed with extracts at 50, 100, 150, 200, 250, and 300 μg/mL. Plate was kept in a dark place for 30 min at room temperature. A Multiskan Sky microplate reader (Thermo Scientific, Waltham, MA, USA) was employed to assess absorbance of each well at 517 nm. For the ABTS assay, a stock solution of ABTS was prepared by dissolving 19.7 mg of ABTS reagent and 186.2 mg of K_2_S_2_O_8_ in 50 mL of distilled water. Mixture was maintained under moderate stirring for 1 h in a dark place. Again, using a 96-well plate, 200 μL of the resultant solution were dispensed, per well, and mixed with extracts at 50, 100, 150, 200, 250, and 300 μg/mL. Exposure to treatment was endured for 6 min, and absorbance was recorded utilizing the same microplate reader. For the H_2_O_2_ assay, 70 μL of a 40 mmol/L H_2_O_2_ solution were mixed with 100 μL of extracts at 50, 100, 150, 200, 250, and 300 μg/mL, and kept in a dark place for 30 min. Given the used volumes, absorbances were assessed in a Cary 60 spectrophotometer (Agilent Technologies, Santa Clara, CA, USA) at 230 nm. Despite experimental differences, ascorbic acid was used as a positive control at the same concentrations as extracts, and Equation (1) was considered to calculate the percentage of scavenged DPPH, ABTS, or H_2_O_2_ radicals. In this equation, *A*_0_ is the absorbance of the DPPH solution, whereas *A*_1_ represents the absorbance of treatment with extracts from *K. tomentosa*. All experiments were performed in triplicate.
(1)$$\%\;Radical\;scavenging\;activity=\frac{\left\{(A_{0}\right.-A_{1}\left.)\right\}}{A_{0}}\times 100$$

Equation (1) determination of the DPPH, ABTS, or H_2_O_2_ scavenging activity of the hexane, chloroform, and methanol extracts from *K. tomentosa*.

### 4.10. Culture of In Vivo Models and Toxicity Assays

The toxicity of extracts from *K. tomentosa* was analyzed against *A. salina* and *C. elegans* as *in vivo* models. Regarding their culture, *A. salina* dried cysts were dispensed in 1 L of distilled water supplemented with 35 g of sea salt, and maintained at 32 °C under vigorous agitation for 48 h. In the case of *C. elegans* nematodes, they were kindly provided by the Department of Chemical Biological Sciences at the Universidad de las Américas Puebla, fed with 600 μL of *E. coli* OP50 strain and maintained in 100 × 15 mm plaques at 22 ± 2 °C. For their synchronization, plaques with adult nematodes were washed with M9 buffer, placed into 2 mL Eppendorf tubes, and centrifuged at 4600 rpm and 4 °C for 1 min. Supernatant was removed, 1 mL of M9 buffer was added to each tube, and nematodes were centrifuged again under the same conditions. After this, supernatant was removed, 1 mL of NaOH 1M was added, and Eppendorf tubes were vortex-shaken (Vortex-Gene 2 G560, Scientific Industries, Bohemia, NY, USA) for 30 s. The supernatant was removed again and 500 µL of NaOH 1M and 500 µL of 1 M NaOH: 5% NaClO were added. Again, Eppendorf tubes were vortex-shaken for 60 s and centrifuged under the same conditions. The supernatant was removed, and the pellet was washed two times with 1 mL of M9 buffer centrifuging at 5600 rpm at 4 °C for 1 min. The pellet containing the eggs was resuspended, seeded in Petri plates previously supplemented with *E. coli* OP50, and incubated for 2.5 days. Toxicity assay against *A. salina* was performed by carefully placing 250 μL of specimens in a 96-well plate together with extracts at 50, 100, 150, 200, 250, and 300 μg/mL. A Leica DMi1 inverted microscope (Leica, Wetzlar, Germany) equipped with a FLEXACAM C1 camera was utilized to monitor nauplii survival rate after 24 h exposure to treatment. Images were acquired through the Leica software version 3.3.0 (Leica Microsystems, Wetzlar, Germany). Comparably, 20 μL of resuspended L4 nematodes were dispensed in a 96-well plate together with 100 μL of M9 buffer containing extracts at 50, 100, 150, 200, 250, and 300 μg/mL. Plate was incubated at 22 ± 2 °C, and the survival rate was monitored for 6 h employing the same invested microscope.

### 4.11. Statistical Analysis

Data from the scavenging activity of extracts from *K. tomentosa* were evaluated by a one-way analysis of variance (ANOVA) followed by Tukey’s mean separation test utilizing OriginPro 2023 data processing software (OriginLab, Northampton, MA, USA).

## Figures and Tables

**Figure 1 pharmaceuticals-17-01051-f001:**
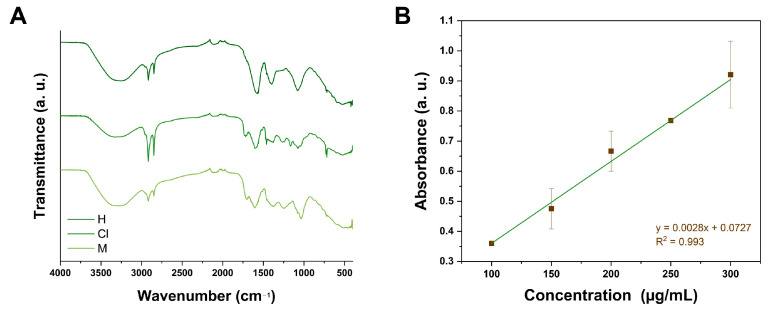
(**A**) FTIR spectra of the hexane (H), chloroform (Cl), and methanol (M) extracts, and (**B**) quercetin calibration curve to estimate TFC.

**Figure 2 pharmaceuticals-17-01051-f002:**
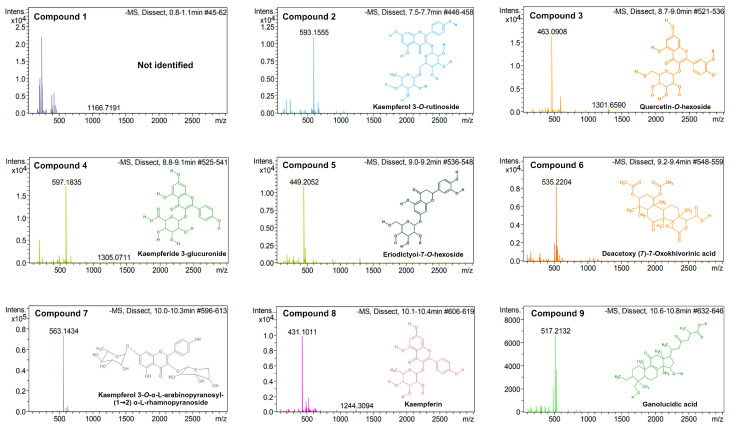
Mass spectra and chemical structures of identified compounds in the methanol extract from *K. tomentosa*: not identified (**1**), kaempferol 3-*O*-rutinoside (**2**), quercetin-*O*-hexoside (**3**), kaempferide-3-glucuronide (**4**), eriodictyol-7-*O*-hexoside (**5**), deacetoxy (7)-7-oxokhivorinic acid (**6**), kaempferol 3-*O*-α-L-arabinopyranosyl-(1→2) α-L-rhamnopyranoside (**7**), kaempferin (**8**), ganolucidic acid C (**9**), kaempferol (**10**), spiraeoside (**11**), apigenin (**12**), linoleic acid (**13**), and heliannuol A (**14**).

**Figure 3 pharmaceuticals-17-01051-f003:**
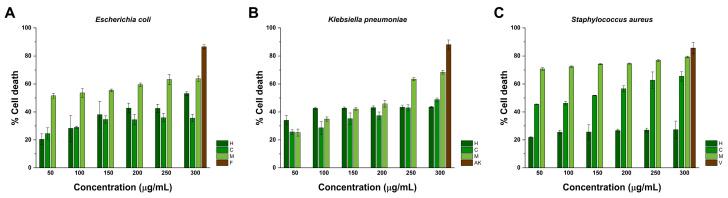
Antibacterial activity of extracts from *K. tomentosa* against (**A**) *E. coli*, (**B**) *K. pneumoniae*, and (**C**) *S. aureus*. Positive controls comprehended fosfomycin (F), amikacin (AK), and vancomycin (V), respectively. The mean ± SD of three independent experiments is shown.

**Figure 4 pharmaceuticals-17-01051-f004:**
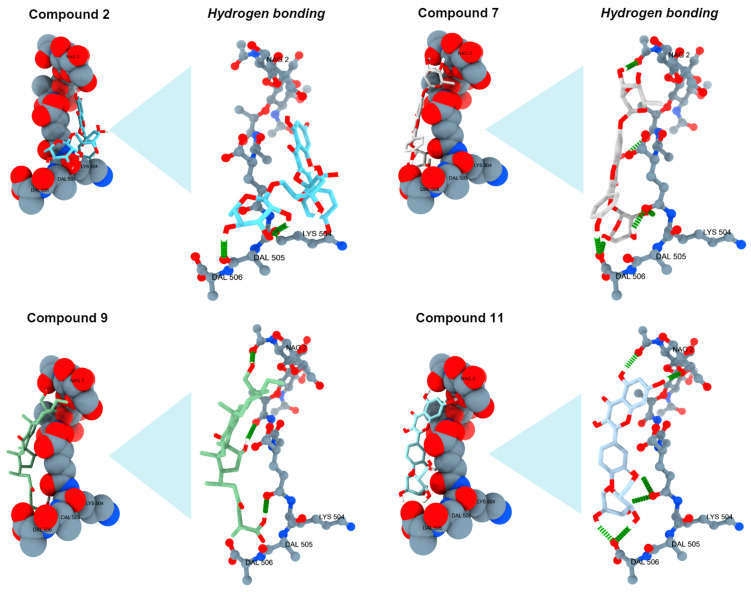
Docking analysis and hydrogen bonding of compounds **2**, **7**, **9**, and **11** with the NAM/NAG-peptide subunits of the *S. aureus* cell wall. Green cylinders represent hydrogen bonds, whereas other colors show elements: oxygen (red), nitrogen (blue), and carbon (ligand).

**Figure 5 pharmaceuticals-17-01051-f005:**
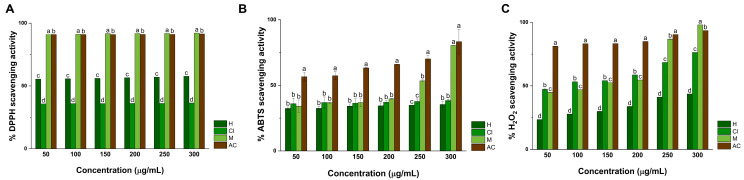
DPPH (**A**), ABTS (**B**), and H_2_O_2_ (**C**) scavenging activity of extracts from *K. tomentosa*. Ascorbic acid (AC) was utilized as a positive control. The mean ± SD of three independent experiments is shown. Different letters indicate significant differences among extracts, which were observed at *p* values significantly below <0.001 evaluated by a one-way ANOVA, followed by a post hoc multiple comparison with Tukey’s test.

**Figure 6 pharmaceuticals-17-01051-f006:**
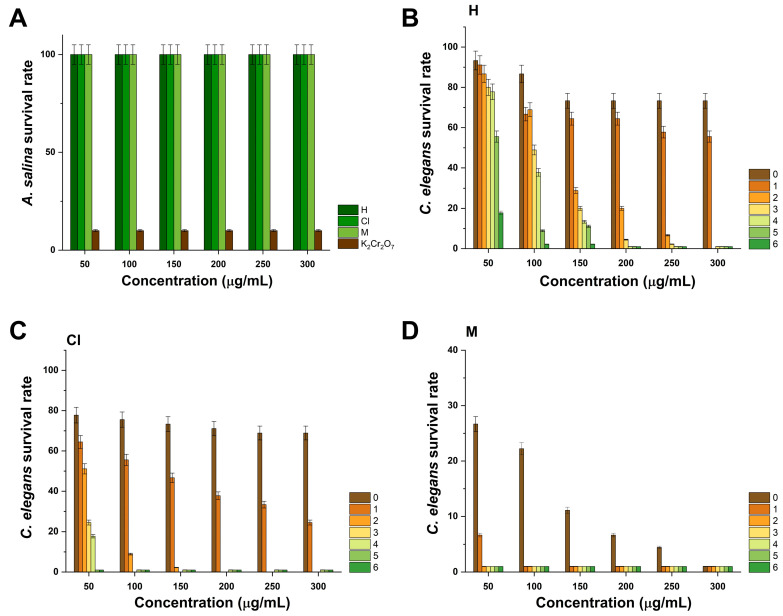
Toxicity of extracts from *K. tomentosa* against (**A**) *A. salina* nauplii and (**B**–**D**) *C. elegans*. Results presented for *A. salina*, and *C. elegans* were obtained after 24 and 6 h of exposure to treatment, respectively.

**Table 1 pharmaceuticals-17-01051-t001:** Compounds identified in the hexane and chloroform extracts from *K. tomentosa* by GC/MS.

Extract	Compound	Rt (min)	Match	R Match
H	β-sitosterol	7.10	774	813
	α-amyrin	8.43	801	811
	β-amirone	8.49	808	821
	α-amyrin	8.61	834	838
	Lupeol	9.14	826	832
	β-amyrin	11.63	825	856
	Urs-12-ene	16.77	780	833
	Lupen-3-one	21.43	815	836
	α-amyrone	21.79	842	851
	Lupenone	23.04	913	915
Cl	β-amyrin acetate	10.67	807	834
	Urs-12-en-24-oic-3-oxo-methyl ester	11.82	801	897
	Octacosane	22.82	898	932
	Heptacosanol	23.63	801	860
	Lupen-3-one	27.77	828	840

Abbreviations: Rt, retention time.

**Table 2 pharmaceuticals-17-01051-t002:** Calculated MIC_90_ values of extracts from *K. tomentosa* towards *E. coli*, *K. pneumoniae*, and *S. aureus*. Concentrations are expressed in μg/mL.

Extract	*E. coli*	*K. pneumoniae*	*S. aureus*
H	609.78	1857.98	3634.67
Cl	7997.28	758.74	574.28
M	776.65	423.59	648.30

Abbreviations: H, hexane; Cl; chloroform; M, methanol.

**Table 3 pharmaceuticals-17-01051-t003:** Binding energies of compounds **2**, **7**, **9**, and **11** with the NAM/NAG-peptide subunits of the *S. aureus* cell wall.

Ligand	Binding Energy (Kcal/mol)	SD	*n*
Compound **2**	−6.1	0.4	9
Compound **7**	−5.9	0.3	20
Compound **9**	−6.3	0.5	35
Compound **11**	−5.7	0.4	29
Vancomycin	−3.1	0.7	11

Abbreviations: SD, standard deviation; *n*, cluster size.

**Table 4 pharmaceuticals-17-01051-t004:** IC_50_ values of hexane (H), chloroform (Cl), methanol (M) extracts from *K. tomentosa*, and ascorbic acid (AC) against DPPH, ABTS, and H_2_O_2_ radicals. Concentrations are expressed in µg/mL.

Extract	DPPH	ABTS	H_2_O_2_
H	2.76	978.14	376.83
Cl	6209.56	1554.73	87.81
M	3.95	210.22	110.75

Abbreviations: H, hexane; Cl; chloroform; M, methanol.

## Data Availability

Data collected during this work can be consulted with authors for correspondence upon reasonable request.

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
