# Peer review of "Kalanchoe tomentosa: Phytochemical Profiling, and Evaluation of Its Biological Activities In Vitro, In Vivo, and In Silico"

_pharmaceuticals, 2024, doi:10.3390/ph17081051_

Round 1

Reviewer 1 Report

Comments and Suggestions for Authors

The manuscript presents an interesting study on the phytochemical profiling and biological activities of Kalanchoe tomentosa (K. tomentosa) extracts. However, several aspects require major revisions for clarity, accuracy and completeness. The following points should be addressed to improve the quality and impact of the paper:

The study investigates the extraction of useful components from K. tomentosa using three different solvents - hexane, chloroform and methanol - and compares the antibacterial and antioxidant activities of these extracts. My main concerns revolve around the extraction characteristics of each solvent and the analysis of the extracted components. The use of GC/MS and UPLC/MS/MS for the qualitative analysis of the extracts is inadequate and requires more careful investigation. In addition, when comparing the antibacterial activity of extracts added to culture media, the solubility and diffusivity of the extracts in the media should be considered, considering the hydrophilicity and lipophilicity of the media.

Specific comments:

Line 10: Please provide a reference for "positive ion electron ionization mass spectrometry". Does this refer to GC/MS?

Figure 1a: Please correct the FTIR legend.

Section 2.3 (GC/MS Results): Add further discussion of the GC/MS results. Explain why the methanol extract was not analyzed by GC/MS. Include axis labels for Figures S1 and S2. Arrange compounds in order of retention time.

Section 2.3 (High boiling compounds): The chloroform extract contains high boiling compounds such as tritetracontane with a boiling point of 540°C. Such compounds are not normally detected by standard GC/MS and may be due to external contamination such as column degradation. Careful consideration should be given to the selection of these compounds.

Figure 5: There appears to be no correlation between extract concentration and antioxidant activity. Explain this observation. Perform statistical analysis to identify significant differences that may clarify the data.

Figure 5 and Table 4 (DPPH antioxidant activity): The methanol extract shows the highest antioxidant activity in Figure 5, followed by hexane and chloroform. However, the IC50 value for DPPH in Table 4 shows that chloroform is significantly higher than the other solvents. Ensure consistency between Figure 5 and Table 4 by carefully re-checking the data.

UPLC/MS/MS analysis: Explain why the hexane and chloroform extracts were not analyzed by UPLC/MS/MS.

FTIR analysis: Compare the FTIR spectra of the original sample and the extraction residue. There is a decrease in transmission below 1000 cm‒1, suggesting the presence of inorganic matter in the extracts. Clarify this observation.

Line 331: Is the peak at 1750‒1735 cm‒1 indicative of fatty acids? Explain why no fatty acids were extracted with hexane.

Figure 1b: Add the detection wavelength for the UV-Vis analysis. Explain why only the methanol extract is shown in Figure 1b.

Line 538: Confirm the column information. Is it the HP-5MS column?

Line 558: Check the flow rate for UPLC/MS/MS. is 8 L/min correct?

Addressing these comments will improve the clarity, accuracy and scientific rigour of the manuscript. Please revise the manuscript accordingly and provide detailed responses to each comment.

Reviewer 2 Report

Comments and Suggestions for Authors

The article proposed by the group of authors is part of the current research areas, the evaluation of new potentially medicinal species from a phytochemical and phytobiological point of view.

The evaluation was based on:

1. The introduction provides information on the role of medicinal plants in health management; systematizes information about K. tomentosa, traditional use in the treatment of infections; studies reporting the effects of different types of plant extracts on pathogens resistant to various other types of treatments; the purpose and objectives of the study are very well specified;

2. The Results section is structured according to the objectives of the study: the content of total flavones on the methanolic extract obtained from K. tomentosa leaves is evaluated, correlated with the FTIR analysis of the extracts in hexane, chloroform and methanol; the GS?MS analysis aimed at the evaluation of lipophilic compounds; the UPLC/MS/MS analysis allowed the identification and evaluation of the content of flavonoid derivatives in the 3 types of extracts; the results from the determination of the antibacterial activity on pathogens resistant to synthetic substances are correlated with the in silico analysis and the determination of the antioxidant action and cytotoxicity tests on invertebrates; all the obtained results are entered in tables, graphically represented by diagrams, structural diagrams; the results are expressed correctly;

3. The discussion section is extended and correlates all the results obtained with data from specialized literature;

4. The Material and methods section is properly structured, the authors describe the protocol for obtaining plant extracts and all types of analyzes to which they are subjected; the analysis conditions are described in detail;

5. I consider the bibliography to be justifiable.
